# Characterization of the Effects of Low-Sodium Salt Substitution on Sensory Quality, Protein Oxidation, and Hydrolysis of Air-Dried Chicken Meat and Its Molecular Mechanisms Based on Tandem Mass Tagging-Labeled Quantitative Proteomics

**DOI:** 10.3390/foods13050737

**Published:** 2024-02-28

**Authors:** Jianhao Li, Zihang Shi, Xiankang Fan, Lihui Du, Qiang Xia, Changyu Zhou, Yangying Sun, Baocai Xu, Daodong Pan

**Affiliations:** 1State Key Laboratory for Managing Biotic and Chemical Threats to the Quality and Safety of Agro-Products, Ningbo University, Ningbo 315211, China; 13484274767@163.com (J.L.); szh11866@163.com (Z.S.); nxiankang2022@163.com (X.F.); dulihui@nbu.edu.cn (L.D.); xiaqiang@nbu.edu.cn (Q.X.); zhouchangyu@nbu.edu.cn (C.Z.); daodongpan@163.com (D.P.); 2Key Laboratory of Animal Protein Food Deep Processing Technology of Zhejiang Province, College of Food Science and Engineering, Ningbo University, Ningbo 315211, China; 3School of Food and Biological Engineering, Hefei University of Technology, Hefei 230601, China; baocaixu@163.com

**Keywords:** salt substitution, protein oxidation and proteolysis, air-dried chicken, proteomics

## Abstract

The effects of low-sodium salt mixture substitution on the sensory quality, protein oxidation, and hydrolysis of air-dried chicken and its molecular mechanisms were investigated based on tandem mass tagging (TMT) quantitative proteomics. The composite salt formulated with 1.6% KCl, 0.8% MgCl_2_, and 5.6% NaCl was found to improve the freshness and texture quality scores. Low-sodium salt mixture substitution significantly decreased the carbonyl content (1.52 nmol/mg), surface hydrophobicity (102.58 μg), and dimeric tyrosine content (2.69 A.U.), and significantly increased the sulfhydryl content (74.46 nmol/mg) and tryptophan fluorescence intensity, suggesting that protein oxidation was inhibited. Furthermore, low-sodium salt mixture substitution significantly increased the protein hydrolysis index (0.067), and cathepsin B and L activities (102.13 U/g and 349.25 U/g), suggesting that protein hydrolysis was facilitated. The correlation results showed that changes in the degree of protein hydrolysis and protein oxidation were closely related to sensory quality. TMT quantitative proteomics indicated that the degradation of myosin and titin as well as changes in the activities of the enzymes, CNDP2, DPP7, ABHD12B, FADH2A, and AASS, were responsible for the changes in the taste quality. In addition, CNDP2, ALDH1A1, and NMNAT1 are key enzymes that reduce protein oxidation. Overall, KCl and MgCl_2_ composite salt substitution is an effective method for producing low-sodium air-dried chicken.

## 1. Introduction

Air-dried poultry products are a traditional delicacy in the Jiangsu and Zhejiang regions of China and are widely liked by consumers for their unique taste. The emergence of hot air drying technology has freed air-dried meat products from the constraints of weather conditions and greatly accelerated the air-drying process [1]. The results of studies on semi-dried Takifugu fillets showed that air-drying at high temperatures resulted in rapid water loss, which was favorable for the sensory qualities of freshness and chewiness but unfavorable for the maintenance of color, odor, and histomorphology, whereas low temperature air-drying showed sensory similarity to natural air-drying and had sensory scores that were only inferior to those of natural air-drying [2]. However, the low-temperature of the air-drying technique requires the use of large amounts of sodium chloride during processing, which is a major concern because excessive sodium can lead to a variety of human diseases. Therefore, there is a need to find a suitable method to reduce the sodium chloride content while further improving the quality of the product.

NaCl is the most commonly used food additive in the preparation of air-dried chicken. Existing studies have found other metal salt substitutions, natural extract substitutions, and physical techniques to reduce salt content without loss of meat quality. For example, replacing 30% of the NaCl with 18% potassium lactate and 12% lysine promoted the oxidation of lipids in dry-cured Kinmen, increased the activities of neutral lipase, acid lipase, and phospholipase, and significantly increased the concentration of free fatty acids and the content of volatile flavor substances [3]. Ultrasonication was a common physical means to improve the physicochemical properties of reconstituted hams with reduction in the NaCl content by 50%. Through ultrasonic treatment, sodium chloride could increase yield, and improve color, and the overall acceptability of taste and texture parameters [4]. However, from the practical application of industrial production and economic perspective, the low-sodium salt mixture substitution strategy still needs further research.

The quality of meat products is influenced by the oxidation and hydrolysis of proteins [5]. Protein oxidation is an important indicator of the flavor and quality of meat. Protein oxidation leads to changes in physicochemical properties such as protein aggregation ability, solubility, emulsification, etc. [6]. The hydrolysis of proteins produces short chains of peptides and free amino acids, which are important factors in the development of the aroma and taste of food [7,8]. While the degree of protein oxidation and hydrolysis is affected by endogenous proteases in meat, the activity of endogenous proteases is affected by salt. Compared to 6% NaCl, 8% NaCl increased the activity of calpain during the curing period, but decreased the cathepsin activity of B, D, H, and L, which explains the softer texture of low-salt Jinhua ham [9]. Different salt substitution strategies also affect endogenous protease activity. KCl partial substitution showed a similar effect to that of 100% NaCl on cathepsin activity. Partial substitution of NaCl with CaCl_2_ and MgCl_2_ had a promoting effect on cathepsin B activity, but had no significant effect on cathepsin H and L activities, while the inhibitory effect of ZnCl_2_ substitution was weak [10]. Currently, proteomics technology is widely used to analyze the quality formation mechanisms of meat products. Proteomics, as the discipline of large-scale analysis of proteins from specific biological systems at a specific time, has contributed greatly to the assessment of the quality, safety, and biological activity of foods. For example, ACTN2, DES, troponin, and heat shock proteins may be the main proteins related to changes in meat quality during low-salt glycerol-mediated curing [11]. In addition, myosin and actin catabolism were verified to be highly responsive to the accumulation of sweet and savory amino acids [12]. Therefore, it is industrially important to investigate the mechanism of low-sodium salt mixture substitution for flavor presentation using TMT quantitative proteomics.

In this study, the effects of low-sodium salt mixture substitution on sensory quality, protein oxidation, and the hydrolysis of air-dried chicken meat and its molecular mechanisms were investigated. Initially, the sensory evaluation was used to determine the effect of a low-sodium salt mixture on air-dried chicken. Furthermore, the effects of low-sodium salt mixture substitution on carbonyl content, sulfhydryl content, surface hydrophobicity, dimeric tyrosine content and tryptophan fluorescence intensity, protein hydrolysis index, and chelating protease activity in air-dried chicken meat were then determined. Finally, the molecular mechanism of flavor presentation was determined via tandem mass tagging quantitative proteomics. This study can provide a theoretical basis for the low-salt air-drying method of compound salt substitution.

## 2. Materials and Methods

### 2.1. Materials

NaCl, KCl, and HCl were purchased from Sinopharm Chemical Reagent Company, (Shanghai, China). Guanidine hydrochloride and DTNB (5,5′-Dithiobis-(2-nitrobenzoic acid)) were purchased from Shanghai Aladdin Biochemical Technology Company, (Shanghai, China). A BCA (Bicinchoninic Acid Assay) kit was purchased from Takara Biotechnology Company (Dalian, China). Tris, urea, and DNPH (2,4-Dimethylphenylhydrazine) were purchased from Beijing Solar Bioscience & Technology Company (Beijing, China). EDTA (Ethylene Diamine Tetraacetic Acid) and glycine were purchased from Rhawn reagent (Shanghai, China). Arg-Arg-AMC and Phe-Arg-AMC were purchased from Nanjing Peptide Biotechnology (Nanjing, China). The remaining reagents were purchased from Shanghai Macklin Biochemical Company (Shanghai, China).

### 2.2. Processing of Air-Dried Chicken

All chickens were purchased from Langde in Ningbo, China. All chickens were gutted and exsanguinated. Then, all the chickens were divided into four treatments, which were marination at 4 °C for 24 h with CK: 8% NaCl; F1: 5.6% NaCl and 2.4% KCl; F2: 5.6% NaCl and 2.4% MgCl_2_; F3: 5.6% NaCl, 1.6% KCl and 0.8% MgCl_2_, respectively (1 g of pickling solution was used per gram of chicken.). After marination, the chickens were drained at room temperature. Immediately afterwards, they were placed in an air-drying oven for processing and air-dried for 3 days at 16 °C and 68% humidity. Then, the air-dried chickens were boiled in water at 100 °C for 0.5 h. All salt ratios were determined by calculating the ratio of the salt mass to the mass of the pickling solution.

### 2.3. Sensory Analysis

Ten consumers (five males and five females, aged 21–25 years) were selected. Sensory evaluations were conducted by a trained 10-member tasting panel. The panel members were trained according to the specifications of the Erick Saldaña et al. with slight modifications [13]. Briefly, samples were packaged in disposable cups and distributed to each member. Between samples, the panelists were instructed to consume water and bread. The finished dried chicken was boiled and sliced, and the bitter taste, astringency, salty taste, fresh flavor, texture, and overall acceptability of the product were evaluated (sensory evaluations are shown in Table 1). The results are presented as the average of ten experiments.

### 2.4. Extraction of Myofibrillar Protein from Air-Dried Chicken

Myofibrillar protein was extracted from air-dried chicken to analyze the effects of different salt substitutions on protein oxidation and hydrolysis. The extraction of myofibrillar protein was based on the prior methodology of the subject group [14]. The 25 g sample was minced, added with 4 times the amount of EDTA-Tris buffer (pH = 8.3), homogenized at 10,000× *g* for 30 s, and centrifuged at 5000× *g* for 20 min. The precipitate was dissolved with 4 times the amount of SSS buffer (20 mM Na_2_HCO_3_/NaH_2_CO_3_, 0.1 M KCl, 2 mM MgCl_2_, and 1 mM EGTA (Ethylene Glycol Tetraacetic Acid), pH = 7), followed by centrifugation at 4000× *g* for 10 min. The pellet was redissolved in 1% Triton X-100 + 99% SSS, and the solution was centrifuged at 4000× *g* for 10 min. After that, the pellet was washed twice with SSS. A solution of 0.1 M KCl was added to the pellet, and the suspension was filtered using four layers of gauze to remove impurities. After centrifugation, 0.1 mM NaCl solution was added to the pellet. Finally, the precipitate was dissolved in twice the volume of distilled water, centrifuged at 5000× *g* for 20 min, and then the supernatant was discarded and the process was repeated twice. The extracted myofibrillar proteins were dissolved in PBS buffer (sodium phosphate buffer pH = 6.5), and the concentration was adjusted to 5 mg/mL using the BCA kit and verified using SDS-PAGE (sodium dodecyl sulfate-polyacrylamide gel electrophoresis).

### 2.5. Determination of the Protein Oxidation Levels in Air-Dried Chicken

#### 2.5.1. Determination of the Myofibrillar Protein Carbonyl Content

The total carbonyl content was measured using DNPH as described by Soglia, Petracci, & Ertbjerg with slight modifications [15]. For each sample, two equal volumes of 1 mL of myofibrillar protein solution (5 mg/mL) were dispensed into 10 mL centrifuge tubes, and 1 mL of 2 M HCl solution was added to the control; then, 1 mL of 2 M HCl solution containing 0.1% DNPH (*v*/*v*) was added to the treatment and the reaction was performed for 1 h (vortexed every 10 min) under protection from light. The proteins were then precipitated with 1 mL of 20% TCA solution (trichloroacetic acid). The resulting sample was then centrifuged at 8000× *g* for 15 min. After removal of the supernatant, the pellet was washed three times with 1 mL of ethanol ethyl acetate (1:1, *v*/*v*), and the supernatant was removed via centrifugation at 8000× *g* for 15 min. Thereafter, the pellet was dissolved in 3 mL of 20 mM PBS buffer with 6 M guanidine hydrochloride. Finally, the solution was centrifuged at 8000× *g* for 5 min and the supernatant was collected to determine the absorbance at 370 nm, and the carbonyl content was calculated using a molar absorption coefficient of 22,000 M^−1^ cm^−1^.

#### 2.5.2. Determination of the Total Sulfhydryl Content of Myofibrillar Proteins

The total sulfhydryl was determined according to the method of Gan et al. with slight modifications [5]. A 1 mL myofibrillar protein solution (5 mg/mL) was mixed with 9 mL of 50 mM PBS buffer as blank control, while the treatment mixed 1 mL of myofibrillar protein solution with 9 mL of Tris-glycine buffer containing 8 M urea, 4 mM EDTA, 0.09 M glycine, and 0.086 M Tris, vortexed and mixed well. Then, 4 mL of the supernatant was placed in a test tube and 0.5 mL of Ellman’s reagent (Tris-glycine buffer with 4 mg/mL DTNB) was added, vortexed, and mixed, and then incubated for 30 min at 25 °C. The absorbance was measured at 412 nm and the sulfhydryl content was calculated using a molar absorption coefficient of 13,600 M^−1^ cm^−1^.

#### 2.5.3. Determination of the Surface Hydrophobicity and Solubility of Myofibrillar Proteins

Surface hydrophobicity was determined by the method used by Chin, Go, & Xiong with slight modifications [16]. A 1 mL myofibrillar protein solution (5 mg/mL) was added to 200 μL of 1 mg/mL bromophenol blue in a centrifuge tube, vortexed (10 min), and centrifuged (8000× *g*, 10 min, 4 °C), and the supernatant was removed and diluted 10 times. The absorption value was measured at a wavelength of 595 nm, and the blank was replaced by PBS without the addition of myofibrillar protein.

Protein solubility was adjusted to 2.5 mg/mL and determined according to the method used by Y. Zhou et al. with slight modifications [17]. The solution was left to stand for 1 h at 4 °C and centrifuged for 15 min (8000× *g* at 4 °C). The supernatant was removed, and the concentration of the supernatant was determined using the bicarbonate method. The ratio of the protein concentration after centrifugation to the protein concentration before centrifugation was the solubility of myofibrillar protein.

#### 2.5.4. Determination of the Dimeric Tyrosine Content of Myofibrillar Proteins

The dimeric tyrosine content was determined by referring to the method used by Gan et al. with slight modifications [5]. Myofibrillar protein was diluted to 1 mg/mL with 20 mM PBS. Insoluble material was removed from the solution via centrifugation at 10,000× *g* for 10 min, the supernatant was collected, and the absorbance value of the protein solution was measured using a fluorescence spectrophotometer at 420 nm and 325 nm for the emission wavelength and excitation wavelength, respectively. The dimeric tyrosine content was equal to the absorbance value divided by the protein concentration in arbitrary units (A.U).

#### 2.5.5. Tryptophan Fluorescence Spectrometry of Myofibrillar Protein

The natural fluorescence of tryptophan was determined via fluorescence spectroscopy using a Hitachi F-4500 fluorescence spectrophotometer (Shimadzu, Japan) as described by Z. Zhu, Mao, Wu, Zhang, & Deng with slight modifications [18]. The natural fluorescence of tryptophan was determined using fluorescence spectroscopy by diluting the myofibrillar protein solution to 0.4 mg/mL in 20 mM PBS. The diluted solution was subjected to intrinsic fluorescence spectroscopy at 300 to 400 nm under the following conditions: excitation wavelength of 283 nm and excitation and emission slit widths of 5 nm.

### 2.6. Determination of Proteolysis during the Processing of Air-Dried Chicken

#### 2.6.1. Determination of P.I. (Proteolysis Index)

P.I. analysis was carried out with reference to the method used by Kęska, Stadnik, Wójciak, & Neffe-Skocińska with slight modifications [19]. The degree of proteolysis was expressed as the P.I. (%), which was the percentage of non-protein nitrogen (NPN) to total nitrogen (TN). TN values were determined using an automatic Kjeldahl nitrogen tester. To analyze the NPN values, 4 g of sample was homogenized sharply with 20 mL of citric acid buffer solution (0.05 mol/L, pH = 6.0) and kept at 4 °C for 2 h. After centrifugation (10,000× *g*, 4 °C) for 15 min, the supernatant was filtered. The total filtrate was mixed with 20 mL of 10% TCA and centrifuged again (5000× *g*, 5 min) at 4 °C overnight. All the filtrates were then digested and analyzed using automatic Kjeldahl determination.

#### 2.6.2. Determination of Cathepsin B and L Activity

Cathepsin B and L activities were determined according to the method described by Zhou, with slight modifications [12]. The muscle samples of cathepsin B and L were extracted and thawed at 4 °C. After thawing, 5 g of chicken was accurately weighed, and 35 mL of 50 mM sodium citrate buffer (including 0.2%Triton X-100 and 1 mM EDTA, pH = 5.0) was added and homogenized with a homogenizer in an ice bath at 12,000× *g* for 10 s each time, and then centrifuged at 10,000× *g* for 20 min at 4 °C. After centrifugation, the supernatant was collected, with the volume made up to 50 mL, and the supernatant was used to determine the activities of cathepsin B and L. The concentration of protein in the supernatant was determined using the BCA protein test kit. According to the method used by Zhao et al. [20], 250 μL of substrate solution (100 μΜ) and 50 μL of enzyme extract were mixed and incubated at 37 °C for 20 min, and the reaction was immediately terminated by the addition of 600 μL of ethanol. The fluorescence intensity was measured at an excitation wavelength of 380 nm and an emission wavelength of 440 nm. An enzyme activity (U) of 1 was defined as the amount of enzyme required to produce 1 nmol of AMC per milligram of protein per min at 37 °C.
enzyme activity (U) = n/m
where n is the amount of AMC substance produced and m is the mass of the protein.

#### 2.6.3. Sodium Dodecyl Sulfate-Polyacrylamide Gel Electrophoresis 

SDS-PAGE was performed to analyze the hydrolysis of salt-soluble proteins according to the method described by Minyi Han et al. with slight modifications [12]. A 5 mg/mL solution of myofibrillar protein was mixed with the loading buffer at a ratio of 1:4 (*v*/*v*) and heated in a metal bath at 100 °C for 10 min. A 5% stacking gel and a 12% separating gel were prepared. The amount of 10 µL of the above sample mixture was loaded into the wells. Molecular weights were determined using standard proteins ranging from 11 kDa to 245 kDa.

### 2.7. Texture Profile Analysis (TPA)

TPA was performed using a TA-XT plus mass analyzer equipped with a cylindrical probe (P/50, 50 mm diameter) that was compressed twice to 40% of its original thickness. Chicken meat was cut into 2 × 2 × 2 cubes. Analysis was performed using the following conditions: pre-speed, 2.00 mm/s; trigger force, 5 g; test speed, 1.00 mm/s; and post speed of 1.00 mm/s. The data acquisition rate was 200 pps [21].

### 2.8. TMT Proteomics

#### 2.8.1. Reversed-Phase Chromatographic Separation

Samples of CK and F3 were further analyzed for proteomics. The total protein from the chicken samples was extracted, and the protein was quantified using a BCA kit. Disulfide bonds were broken using dithiothreitol at a final concentration of 5 mM, and 10 mM iodoacetamide was added to the protein solution to modify the SH group. The protein solution was precipitated by adding 6 volumes of cold acetone and then centrifuged. The precipitate was reconstituted by adding 100 μL of 200 mM TEAB (Triethylammonium bicarbonate), and 1 mg/mL Trypsin-TPCK (N-p-Tosyl-L-phenylalanine chloromethyl ketone) was added to 1/50 of the sample mass and digested overnight at 37 °C. The digested peptides were labeled using a TMT labeling kit [12]. The model of liquid chromatography was the Agilent 1100 HPLC column: Agilent Zorbax extend–C18 narrow-diameter column, 2.1 × 150 mm, 5 μM detection wavelength: ultraviolet 210 nm. Mobile phase A: ACN-H_2_O (2:98, *v*/*v*), and the pH was adjusted to 10 with ammonia water. Mobile phase B: ACN-H_2_O (90: 10, *v*/*v*), and the pH was adjusted to 10 with ammonia water. Flow rate: 300 μL/min; Gradient elution conditions: 0~8 min, 98% A; 8~8.01 min, 98–95% A; 8.01~30 min, 95–80% A; 30–43 min, 80–65% A; 43–53 min, 65–55% A; 53–53.01 min, 55–10% A; 53.01–63 min, 10% A; 63–63.01 min, 10–98% A; 63.01–68 min, 98% A. After collection, it was evaporated, and the samples were frozen and stored for mass spectrometry.

#### 2.8.2. Chromatographic and Mass Spectrometry Conditions

The sample was loaded into the analytical column Acclaim pep map RSLC at the flow rate of 300 nL/min and separated at 75 μm × 50 cm (RP-C18, Thermo Fisher (Shanghai, China)). Mobile phase A: ACN-H_2_O-FA (99.9:0.1, *v*/*v*/*v*); Mobile phase B: ACN-H_2_O-FA (80:19.9:0.1, *v*/*v*/*v*); Gradient elution conditions: 0–4 min, 8–11% B; 4–36 min, 11–45% B; 36–39 min, 45–100% B; 39–45 min, 100% B.

The mass resolution of the first-class MS was set to 4.5× 10^4^, the automatic gain control value was set to 3× 10^6^, and the maximum injection time was 30 ms. The mass spectrum scanning was set to the full scanning charge-to-mass ratio *m*/*z* range of 350–1500 *m*/*z*, and 20 of the highest peaks were scanned by MS/MS. All MS/MS spectra were collected via high-energy collisional cracking in the data-dependent positive ion mode, and the collision energy was set to 32. The resolution of MS/MS was set to 3000, the automatic gain control was set to 2 × 10^5^, and the maximum ion injection time was 40 ms. The dynamic exclusion time was set to 30 s. The protein–protein interaction was analyzed by using the string (https://string-db.org) in the Sus Scrofa database.

### 2.9. Statistical Analysis

Each experiment was repeated three times independently, and the results are presented as mean ± standard deviation (SD). The one-way analysis of variance (ANOVA) function in SPSS software (SPSS 25, SPSS Inc., Chicago, IL, USA) was used to analyze the experimental data. A *p*-value < 0.05 was statistically significant. Graphs of the experimental data were generated using Origin 2022 software (Origin Lab Corp., Northampton, MA, USA). 

## 3. Results and Discussion

### 3.1. Sensory Evaluation 

The results of the sensory evaluations are shown in Figure 1. Compared to CK, F2 showed a decrease in all indices, which might be attributed to the fact that divalent cations were characterized by the production of bitter, metallic, astringent, and irritating tastes, and therefore the acceptability of the final product was lower, which is consistent with the findings of existing reports [22]. F1 showed a slight change in both bitterness and off-flavor scores, but the difference was not significant (*p* < 0.05); saltiness, freshness, texture, and overall acceptability scores were lower. Zhixin Gao et al. found that the main reason for the poorer overall acceptability in the 60% NaCl + 40% KCl treatment was due to the lower hardness and bitterness of the resulting product [23]. While José M. Lorenzo et al. found that 50% KCl substitution resulted in severe bitter taste defects and lack of salty taste [24], F3 showed no significant decrease in bitterness, off-flavor, saltiness, and overall acceptability values as well as an increase in freshness and textural qualities, which could be attributed to the reduction in the amount of each salt by different salt substitutions, thus avoiding sensory defects due to the over-substitution of one salt.

### 3.2. Analysis of the Total Carbonyl Content in Different Treatments

The high level of protein oxidation in protein can adversely affect food quality, and consumption of protein may lead to a range of hazards and diseases [25]. The main formation mechanism is that NH- and NH_2_- on the side chain of protein amino acids (mainly including lysine, proline, and arginine) are directly oxidized by hydroxyl radicals, and then oxidized into carbonyl treatments [26]. Therefore, the change in carbonyl content can reflect the oxidation level of the protein. The change in total carbonyl content of air-dried chicken after three days of air-drying is shown in Figure 2. The carbonyl content of CK was 2.66 nmol/mg. The carbonyl content in F1 and F2 was significantly higher (3.94 and 5.00 nmol/mg, respectively) than that in the control. In F3, the carbonyl content was significantly decreased with a value of 1.52 nmol/mg. It was found that 30%, 50%, and 70% KCl instead of NaCl significantly increased the carbonyl content in protein, which was consistent with the results in another work [5]. In addition, another study found that the compound substitution of KCl and MgCl_2_ reduced the carbonyl content [27]. This may be because K^+^ increases the surface tension of proteins and promotes a more relaxed protein structure, which in turn exposed the amino acids to oxidation, leading to an increase in carbonyl content [28]. In addition, MgCl_2_ has a higher Cl^−^ content compared to NaCl, which enhances electrostatic repulsion and swells the structure, thus promoting protein oxidation.

### 3.3. Analysis of the Total Sulfhydryl Content in Different Treatments

The sulfhydryl group of cysteine is very sensitive to the oxidation of ROS, resulting in the formation of various oxidation products and the formation of disulfide crosslinking [29]. The effect of different treatments on the sulfhydryl content of air-dried chicken is shown in Figure 2B. The sulfhydryl content in F2 was 54.81 nmol/mg, which was significantly lower than that in CK (*p* < 0.05), and reached 47.22 nmol/mg; F1 and F3 significantly increased the sulfhydryl content of myofibrillar protein (*p* < 0.05), namely 60.95 and 74.46 nmol/mg, respectively. The sulfhydryl results were generally consistent with the trend for carbonyl, with slight differences in the KCl replacement treatment. A similar conclusion to the one in that study was also found in the present research [27,30]. The research results of Gan et al. showed that after bacon was made, all KCl substitution treatments increased the total sulfhydryl content, but the difference was not significant.

In the process of oxidation, the modification of the amino acid side chain (e.g., loss of mercaptan or formation of carbonyl) is the most important change affecting the quality of muscle food. On the other hand, in this study, it was found that the carbonyl content of F3 decreased, and the sulfhydryl content increased as compared to the control, which in one way can explain the improvement in the texture quality of F3.

### 3.4. Analysis of Surface Hydrophobicity and Protein Solubility in Different Treatments

Protein oxidation can affect surface hydrophobicity because oxidation leads to denaturation and unfolding of protein, resulting in more hydrophobic groups being exposed and increasing surface hydrophobicity [31]. Therefore, the hydrophobic force was determined. As can be seen from Figure 2C, the surface hydrophobicity of CK was 112.38 μg. Compared with CK, the surface hydrophobicity of myofibrillar protein was significantly reduced (*p* < 0.05) by F1 and F2 with the values of 117.10 and 140.63 μg, respectively, and the surface hydrophobicity of F3 was significantly increased with a value of 74.46 μg (*p* < 0.05). It is speculated that the substitution of KCl and MgCl_2_ compound salt might reduce the degree of protein oxidation, and then reduce the surface hydrophobicity. Different kinds and concentrations of metal cations may lead to different surface hydrophobicity. Ca^2+^ and Mg^2+^ ions can reduce the free energy required to transfer nonpolar groups into water, thus promoting the development of protein and the exposure of hydrophobic groups [32]. K^+^ promotes changes in protein structure while reducing the hydrophobic core, resulting in protein separation and susceptibility to oxidation during drying, exposing more hydrophobic groups [30]. 

The solubility of proteins is greatly influenced by protein oxidation and surface hydrophobicity. As shown in Figure 2D, the solubility of CK treatment was 0.68. Compared with CK, the solubility of protein was reduced in all three treatments (*p* < 0.05), the values of F1, F2, and F3 being 0.61, 0.29, and 0.52, respectively. Compared with NaCl and KCl, MgCl_2_ affects the dissolution of proteins [33], which may be the reason for the slight difference between solubility and surface hydrophobicity. 

### 3.5. Analysis of Dimeric Tyrosine Content with Different Salt Substitutions

Protein oxidation promotes the formation of dimeric tyrosine from tyrosine residues. Therefore, the content of dimeric tyrosine may also reflect the degree of protein oxidation. The dimeric tyrosine content of air-dried chicken prepared using different treatments is shown in Figure 2E. The dimeric tyrosine content of CK was 3.07 A.U. F1 increased the content of dimeric tyrosine, but the difference was not significant (*p* > 0.05), F2 significantly increased the content of dimeric tyrosine with a value of 3.40 A.U. (*p* < 0.05), and F3 significantly reduced the content of dimeric tyrosine with a value of 2.69 A.U. (*p* < 0.05). The existing research also showed that as the degree of oxidation increases, the content of dimeric tyrosine increases significantly, which is consistent with the result of this work [34,35]. The results of dimeric tyrosine were consistent with those of the carbonyl group, suggesting that F3 reduces protein oxidation.

### 3.6. Analysis of Tryptophan Fluorescence in Different Treatments

The direct oxidative hydrolysis of tryptophan and protein unfolding result in the loss of fluorescence intensity when tryptophan is exposed to solvents [36]. Therefore, the oxidation level of myofibrillar protein can be determined by measuring the fluorescence intensity of tryptophan. As shown in Figure 2F, the fluorescence intensity of F3 was the highest, followed by that of CK, while the tryptophan fluorescence intensity of myofibrillar protein was significantly reduced by F1 and F2, respectively, and the fluorescence intensity of F2 was the lowest. Compared with NaCl, KCl and MgCl_2_ both caused the protein to “salt in”, thus achieving the effect of unfolding and exposing the internal hidden aromatic amino acid residues (tyrosine, phenylalanine, tryptophan), and divalent salts have a better effect [37]. These exposed amino acid residues were susceptible to oxidation during the air-drying process, resulting in a reduction in tryptophan content.

The results of the carbonyl group, sulfhydryl group, surface hydrophobicity, solubility, dimer tyrosine content, and tryptophan fluorescence spectroscopy showed that the use of KCl and MgCl_2_ reduced protein oxidation. 

### 3.7. Analysis of the Proteolysis Index in Different Treatments

The proteolysis index is widely regarded as a characterization of proteolysis intensity. Figure 3A shows the effect of the partial substitution of different metal ion salts on the proteolysis of air-dried chicken. The P.I. value of CK was 0.059. Compared with CK, the P.I. value of the treatments was higher (*p* < 0.05), its values being 0.072, 0.081, and 0.067, respectively. The substitution of different compound salts promoted the hydrolysis of protein. Proteolysis led to an increase in non-protein nitrogen, which might be an important source of taste compounds mainly involved in free amino acids and oligopeptides [38]. However, the dramatic increase in P.I. might lead to high adherence and a strong bitter flavor in dry-cured meat products [39]. This well explains the severe bitterness and astringency of treatments F1 and F2, and the improvement of bitterness, astringency, and umami quality in F3. Proteolysis promotes the taste of dry-cured ham through the decomposition of muscle structure [38]. One of the key factors affecting proteolysis is the existence of endogenous enzymes, such as calpain and cathepsin [10]. Therefore, the enzyme activity of samples treated with different salt substitutions was further determined.

### 3.8. Analysis of Cathepsin B and L Activity, SDS-PAGE, and Texture in Different Treatments

Cathepsin B and L are closely related to proteolysis. Figure 3B shows the changes in cathepsin activity in air-dried chicken with different salt substitutions. The cathepsin B and L activity of CK were 79.57 U/g and 179.86 U/g, respectively. Compared with CK, the activities of cathepsin B and L activity in all treatments were significantly increased (*p* < 0.05). The cathepsin B activity of F1, F2, and F3 were 103.81 U/g, 112.27 U/g, and 102.13 U/g, respectively. The cathepsin L activity of F1, F2, and F3 were 441.53 U/g, 572.44 U/g, and 349.25 U/g, respectively. It was found that NaCl substituted with KCl, CaCl_2_, and MgCl_2_, respectively, could significantly improve the activity of cathepsin B, which is consistent with the conclusion in this work [10]. The role of cathepsin is essential for the development of flavor in meat products during air-drying processes. Cathepsin activity is responsible for the formation of polypeptides, which are then degraded by aminopeptidases to oligopeptides and free amino acids, thus helping to improve the flavor of meat products [12]. This can be used to explain the improvement in flavor via F3. In addition, the hydrolysis of myofibrillar protein by endogenous enzymes will also affect the texture and tenderness of meat products. David S. Dang et al. activated calpain by inhibiting the mitochondrial calcium one-way transporter, which confirmed that the texture and tenderness of meat products are affected by calpain [40].

SDS-PAGE was used to investigate the hydrolysis of proteins. The characteristic bands of myofibrillar mainly included the myosin heavy chain (MHC, 200 kDa), α-actin (75 kDa), actin (42 kDa), Tropomyosin (Tm, tropomyosin, 34–36 kDa), and troponin-T (37 kDa). It can be seen from Figure 3C that different low-sodium salt mixture substitution had different effects on electrophoresis bands. Compared with CK, F1 showed similar properties on bands, which was reported in previous studies [41]. However, the brightness of the myosin heavy chain, actin, α-actin, troponin-T, and tropomyosin bands was lower in F2 and F3. There are two reasons for the reduced brightness of the protein bands: on one hand, protein oxidation can lead to protein cross-linking and aggregation through disulfide bonds, thus forming aggregates larger than 245 kDa, which are all gathered at the top of the concentrated gel and cannot enter the separation gel, leading to a decrease in the brightness of the protein bands. On the other hand, proteolysis is also an important factor in reducing the brightness of stripes [42,43,44]. The hydrolysis of protein was more severe, especially in the myosin heavy chain, troponin-T, and myosin light chain, which could be attributed to the high residual activity of cathepsins B and L in modern processing [38].

The higher the degree of proteolysis, the softer the texture of the meat, so the texture tests were performed. The texture results are shown in Table 2. Compared with the CK treatment, the hardness, and chewiness of the samples from the three treatments decreased significantly, which might be due to their higher tissue protease activity, hydrolysis of the protein and changes in the network structure of the meat, thus resulting in a softer texture. However, the high protease activity in F1 and F2 severely damaged the texture of the meat and reduced the taste of the meat, so their sensory scores were lower. M.M. Schmidt et al. showed a significant (*p* < 0.05) reduction in the hardness and chewiness of meat products by composite substitution of KCl, MgCl_2_, and CaCl_2_ [45]. It has also been shown that KCl substitution for NaCl reduced the hardness of meat products, which is consistent with the results of this paper [46].

### 3.9. Relationship between Protein Oxidation and Protein Proteolysis

Pearson correlation analysis was carried out to determine whether protein oxidation affects proteolysis in the production process. Its result is shown in Figure 3D. Firstly, there were significant correlations among carbonyl content, sulfhydryl content, surface hydrophobicity, and dimeric tyrosine in protein (*p* < 0.001). The correlation between P.I., cathepsin L, and cathepsin B was significant (*p* < 0.001), which indicated a better correlation between protein oxidation indicators and hydrolysis indicators. P.I. was positively correlated with total carbonyl content (β = 0.733, *p* < 0.001), surface hydrophobicity (β = 0.762, *p* < 0.001), solubility (β = −0.822, *p* < 0.05), bitterness (β = −0.586, *p* < 0.05), Odor (β = −0.810, *p* < 0.01), salty taste (β = −0.533, *p* < 0.05), texture (β = −0.587, *p* < 0.05), and overall acceptability (β = −0.618, *p* < 0.05). In addition, there was a correlation among protein oxidation, hydrolysis, and sensory qualities, which indicated that protein hydrolysis is an important factor affecting the sensory qualities of air-dried chicken. In the existing reports, during the processing of meat products, they were easily influenced by the free radical oxidation system, resulting in protein carbonylation and crosslinking [47], thus forming carbonyl protein. Meanwhile, these formed carbonyl protein solutions were hydrolyzed by endogenous enzymes [5]. The reason may be that protein oxidation changed the higher-order structure of protein and the spatial conformation of the protein, resulting in the unfolding of the protein, which is beneficial to the recognition and hydrolysis of protein by protease [48]. However, the protein oxidation and proteolysis results of F3 showed that with the decrease of protein oxidation level, the level of proteolysis increased, which may be because interactions between KCl and MgCl_2_ reduce protein cross-linking in the oxidation environment, thus reducing protein aggregation, protein cross-linking, and the resistance of aggregates to enzymatic hydrolysis. Protein oxidation reduced proteolysis during the in vitro digestion of pork and beef patties [49].

### 3.10. Proteomics Analysis 

Proteomics experiments were conducted to reveal the mechanism behind the effect of different salt substitution treatments on the quality of air-dried chicken. F3 was selected as the best sensory quality in the treatment, and the control was CK. As shown in Figure 4A, the LC-MS/MS analysis produced 341,908 spectra, of which a total of 27,634 peptides and 3721 proteins were identified (FDR ≤ 0.01). Principal component analysis showed that there was a significant difference between CK and F3 (Figure 4B).

Fold changes > 1.5 or <0.67 at *p* < 0.05 were used to identify differential proteins linked to the proteomic changes in air-dried chicken between treatment B and control A. The volcano plot reveals that a total of 101 proteins were significantly different between F3 and CK, including 56 up-regulated proteins and 45 down-regulated proteins (Figure 4C). Among them, 28 proteins (shown in Table 3) were closely related to this study. The results of the cluster analysis heat map showed good clustering between CK and F3, as shown in Figure 4D.

Analysis of the differential proteins revealed that the myosin light chain 1, Titin isoform b11, and TTN, which are components of myofibrillar fiber proteins, were significantly reduced. The hydrolysis of myosin and actin typically result in the production of large amounts of peptides and free amino acids, which are key substances contributing to the intense and pleasant flavor of freshness, bitterness, sweetness, and richness of meat products. Several peptides derived from the hydrolysis of troponin, myosin, and titin, including GAG, APPPPAEVHEV, EA, and EE, have been described as permanent taste-active compounds in fermented meat products [12]. This might account for the effect of 20% KCl and 10% MgCl_2_ substitution on the flavor quality of air-dried chicken. AKR1E2 and SQRDL are oxidoreductases, as indicated by their altered expression; GSTA4 is the enzyme in glutathione synthesis, glutathione is an important antioxidant in organisms, and the increase in its content could explain the inhibition of protein oxidation in the treatment; ALDH1A1 is an enzyme of retinol metabolism, which can resist oxidation from the environment by down-oxidizing retinol to retinoic acid. Furthermore, NMNAT1 is a protein in the PPAR pathway, and the PPAR pathway has the function of anti-oxidative stress, while the increase in its content could also explain the inhibition of oxidation in the treatment. CNDP2 is an enzyme that catalyzes the degradation of myostatin, an antioxidant, and sweet-tasting dipeptide. DPP7 and ABHD12B contain serine protease activity that degrades proline-containing proteins, resulting in proline-containing dipeptides. Since proline is a hydrophobic amino acid, the dipeptides that it forms with other amino acids tend to have a bitter flavor. FADH2A is an enzyme that degrades phenylalanine and tyrosine, and AASS is an enzyme that degrades lysine. Changes in the levels of these enzymes might be responsible for the altered qualities of bitter, salty, and sweet flavors in the F3 treatment.

Proteins do not usually work alone but interact with other proteins to perform various functions. Therefore, differential protein interaction network analysis was carried out to elucidate the mechanism behind differential protein interactions. Protein–protein interactions were analyzed using STRING as shown in Figure 5, with a *p*-value of 3.65 × 10^−10^ and a clustering coefficient of 0.658 for protein–protein interaction enrichment. The results of the PPI network showed a close relationship between proteins such as AASS, CTH, AKR1E2S, ALDH1A1, and CNDP2 suggesting that proteins which affect the protein oxidation level closely interact with each other, whereas there were also interactions found between FAHD2A and HYKK, suggesting that proteins which affect the proteolysis degree are also interconnected. All the above interactions between proteins were found to be close, which indicates that there is an interaction between protein oxidation and hydrolysis, which matches the experimental results mentioned above. In addition, there were also interactions found between the proteins, MYOZ3, TRIM63, MYLK2, MYL1, and SYNPO2, suggesting that there are interactions between myosin and its connecting proteins.

The proteomic results show that the partial substitution of KCl and MgCl_2_ for NaCl could inhibit protein oxidation and promote hydrolysis, and the interaction between protein oxidation and hydrolysis was also well illustrated by the PPI interaction network, which is consistent with the findings of protein oxidation index and hydrolysis index.

## 4. Conclusions

The effects of low-sodium salt mixture substitution on sensory quality, protein oxidation, and the hydrolysis of air-dried chicken meat and its molecular mechanisms were investigated based on TMT quantitative proteomics. A composite salt formulated with 1.6% KCl, 0.8% MgCl_2_, and 5.6% NaCl could replace 30% of NaCl, improving the freshness and texture quality scores in the air-dried chicken sensory evaluation. It significantly decreased the carbonyl content (1.52 nmol/mg), surface hydrophobicity (102.58 μg), and dimeric tyrosine content (2.69 A.U.), and significantly increased the sulfhydryl content (74.46 nmol/mg) and tryptophan fluorescence intensity in air-dried chicken, suggesting that protein oxidation was inhibited. In addition, the protein hydrolysis index (0.067) and cathepsin B and L activities (102.13 U/g and 349.25 U/g) were significantly increased, suggesting the promotion of protein hydrolysis. The changes in the degree of protein hydrolysis and protein oxidation were found to be closely related to sensory quality. TMT quantitative proteomics indicated that the degradation of myosin and titin, as well as changes in enzyme activities such as CNDP2, DPP7, and ABHD12B, FADH2A, and AASS, were responsible for the change in the taste quality of air-dried chicken via KCl and MgCl_2_ compound substitution. Furthermore, CNDP2, ALDH1A1, and NMNAT1 were found to be key enzymes that reduce protein oxidation. This is of strategic importance for the development of low-sodium salt mixtures for air-dried chicken and the elucidation of their flavor mechanisms.

## Figures and Tables

**Figure 1 foods-13-00737-f001:**
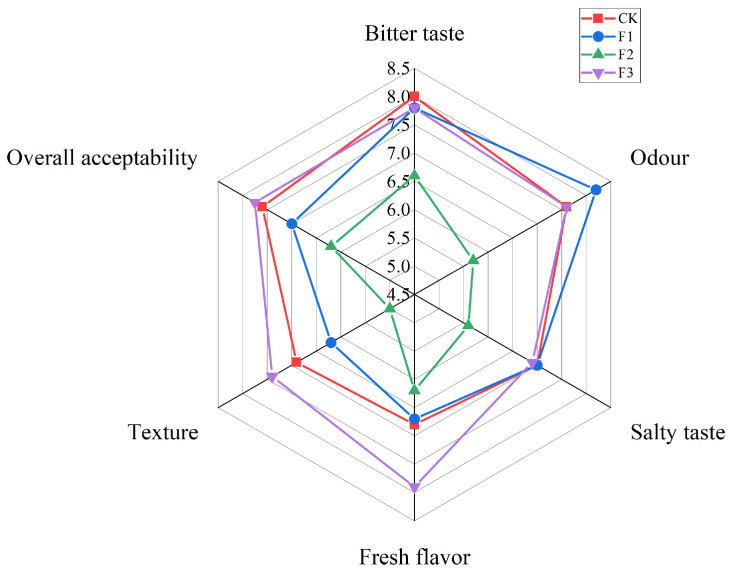
Sensory evaluation results of air-dried chicken with different salt substitution.

**Figure 2 foods-13-00737-f002:**
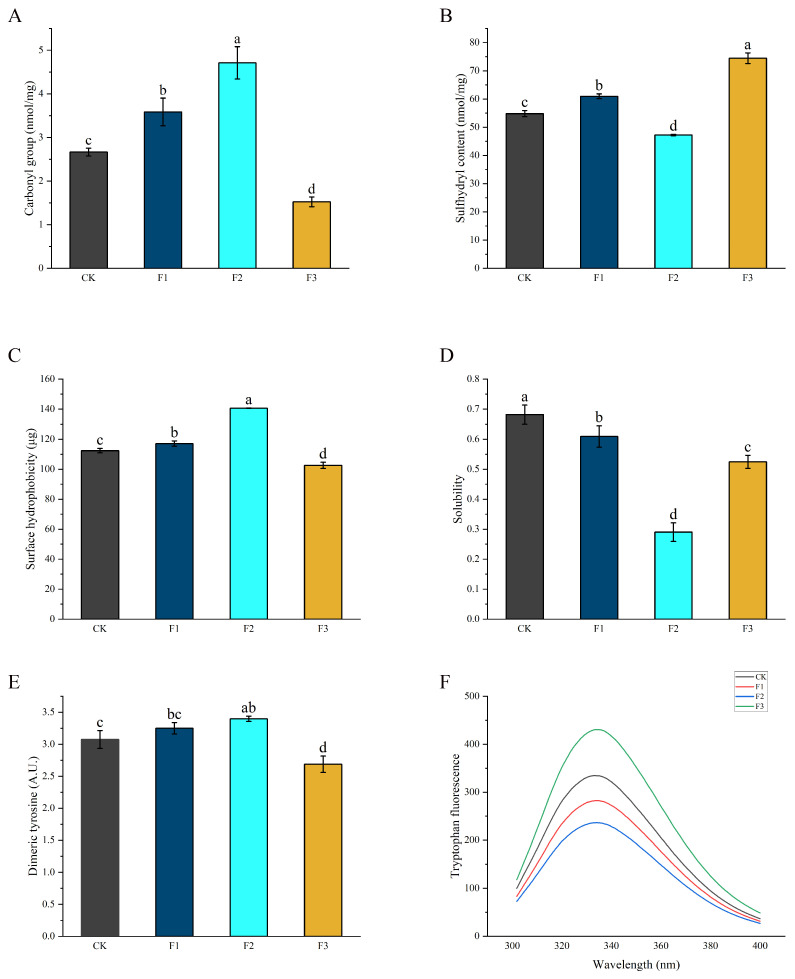
The effect of different salt substitution ratios on the degree of protein oxidation in air-dried chicken. (**A**) Total carbonyl content; (**B**) Total sulfhydryl content; (**C**) Surface hydrophobicity; (**D**) Protein solubility; (**E**) Dimeric tyrosine content; (**F**) Tryptophan fluorescence. a, b, c, and, d represent a significant difference between treatments. CK, F1, F2, F3 represent different treatment conditions. CK: 8% NaCl, F1: 5.6% NaCl + 2.4% KCl, F2: 5.6% NaCl + 2.4% MgCl_2_, F3: 5.6% NaCl + 1.6% KCl + 0.8% MgCl_2_. *p* < 0.05 was considered a significant difference.

**Figure 3 foods-13-00737-f003:**
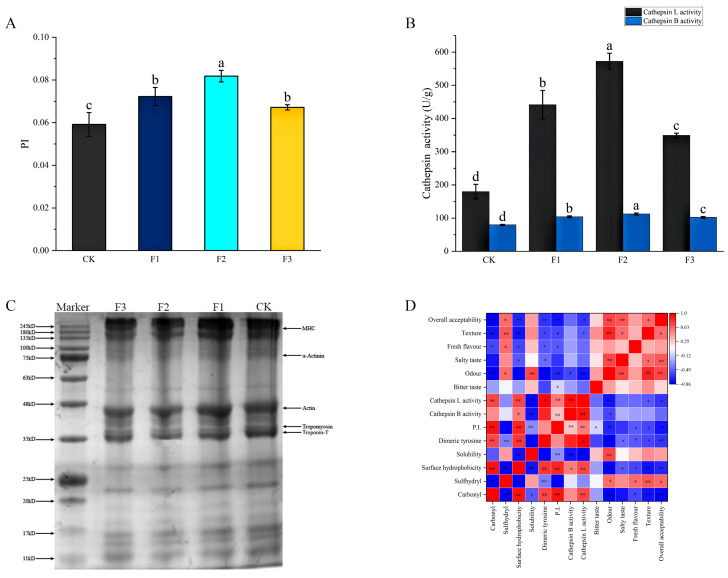
The effect of different treatment conditions on the degree of protein hydrolysis and correlation analysis result between protein oxidation and hydrolysis. (**A**): Results of P.I., (**B**): Results of cathepsin B and L, (**C**): Results of SDS-PAGE, (**D**): Correlation analysis result of protein oxidation, hydrolysis, and sensory quality. The redder the color, the higher the degree of positive correlation, and the bluer the color, the higher the degree of negative correlation. ** represents highly significant correlation (*p* < 0.01), * represents significant difference in correlation (*p* < 0.05). a, b, c, d represent a significant difference between treatments. CK, F1, F2, F3 represent different treatment conditions. CK: 8% NaCl, F1: 5.6% NaCl + 2.4% KCl, F2: 5.6% NaCl + 2.4% MgCl_2_, F3: 5.6% NaCl + 1.6%KCl + 0.8% MgCl_2_. *p* < 0.05 was considered a significant difference.

**Figure 4 foods-13-00737-f004:**
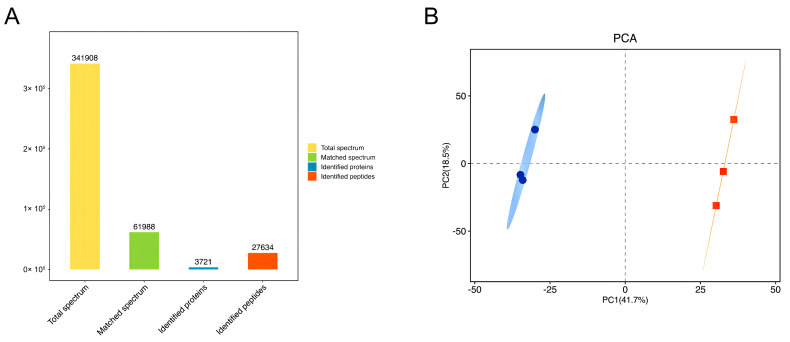
Results of TMT proteomics. (**A**) All identified proteins. (**B**) Results of PCA analysis. Blue color indicates CK treatment; red color indicates F3 treatment. (**C**) The volcanic map of differential proteins; red represents significantly increased proteins; light red represents FC < 1.5; dark red represents FC > 1.5; blue represents significantly decreased proteins; light blue represents FC > 0.67, dark blue represents FC < 0.67; gray represents non-significantly different proteins. (**D**) Heatmap of differential protein clustering analysis.

**Figure 5 foods-13-00737-f005:**
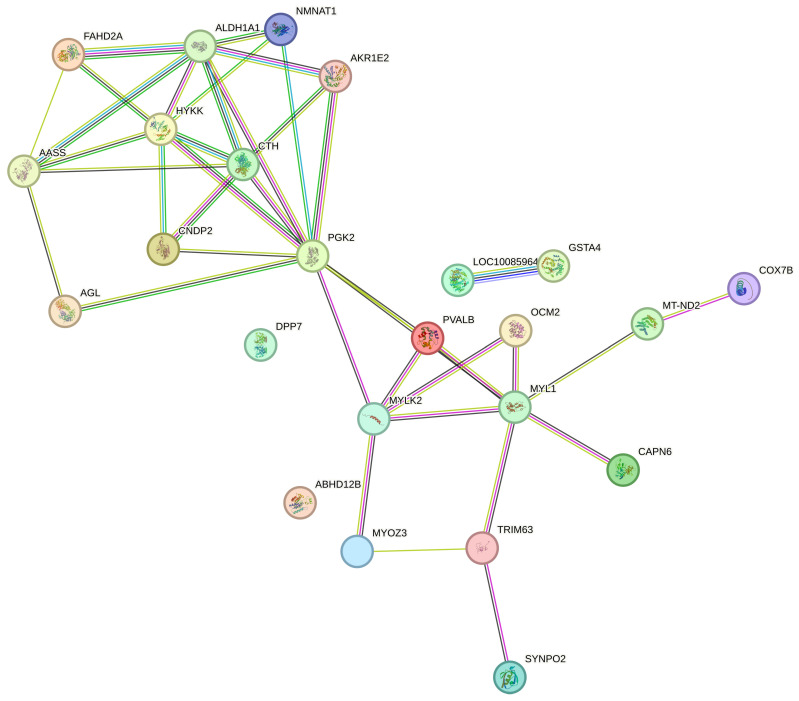
Interaction network results for different proteins.

**Table 1 foods-13-00737-t001:** Scoring criteria for sensory evaluation.

Sensory	Bitter Taste	Astringent	Salty Taste	Fresh Flavor	Texture	Overall Acceptability
7–9	Litter bitter	Little astringent	Moderate saltiness	Moderate flavor	Moderate texture	Very acceptable
4–6	Bitter	Astringent	Saltier or lighter	Fresher or lighter	Softer or harder	acceptable
1–3	Very bitter	Very astringent	Too salty or litter salty	Too fresh or litter fresh	Too soft or too hard	inacceptable

**Table 2 foods-13-00737-t002:** Texture of air-dried chicken with different salt substitution.

	Hardness	Springiness	Cohesiveness	Gumminess	Chewiness	Resilience
CK	5041 ± 410 ^a^	0.835 ± 0.087 ^a^	0.852 ± 0.034 ^a^	4307 ± 483 ^a^	3556 ± 129 ^a^	0.693 ± 0.0261 ^a^
F1	3538 ± 280 ^b^	0.770 ± 0.019 ^a^	0.755 ± 0.034 ^bc^	2659 ± 208 ^bc^	2039 ± 136 ^b^	0.576 ± 0.0387 ^b^
F2	2768 ±251 ^b^	0.826 ± 0.020 ^a^	0.782 ± 0.007 ^b^	2166 ± 214 ^c^	1785 ± 187 ^b^	0.537 ± 0.0380 ^b^
F3	4846 ± 479 ^a^	0.810 ± 0.046 ^a^	0.714 ± 0.026 ^c^	3885 ± 470 ^ab^	2862 ± 543 ^a^	0.563 ± 0.0382 ^b^

a, b, c represent a significant difference between treatments.

**Table 3 foods-13-00737-t003:** Differentially expressed proteins identified in TMT proteomics.

Accession	Gene Name	Description
F1NX83	AGL	Glycogen debranching enzyme
F1P463	CNDP2	CNDP dipeptidase 2 (metallopeptidase M20 family)
A0A8V0ZLR4	HSPB9	Heat shock protein family B (small) member 9
F1NU17	PGK2	Phosphoglycerate kinase
C1L370	pvalb1	Parvalbumin
P02604		Myosin light chain 1, skeletal muscle isoform
E1BYF1	CTH	Cystathionine gamma-lyase
A6BLM8	TTN	Titin isoform b11 (Fragment)
A0A8V0XA66	AASS	Aminoadipate-semialdehyde synthase
F1NMM5	ABHD12B	Abhydrolase domain containing 12B
P79757		Connectin/titin (Fragment)
A0A8V0YKB8	AKR1E2	Aldo-keto reductase family 1 member E2
P19753		Parvalbumin, thymic
A0A343CVV5	ND2	NADH-ubiquinone oxidoreductase chain 2
A0A8V0ZDH7	FAHD2A	Fumarylacetoacetate hydrolase domain containing 2A
A0A8V0Z2L5	TRIM63	Tripartite motif containing 63
A0A8V1A9H4	LDHD	Lactate dehydrogenase D
E1C5M4	HYKK	Hydroxylysine kinase
F1NL83	COX7B	Cytochrome c oxidase subunit 7B
A0A8V0XM57	SYNPO2	Synaptopodin 2
A0A8V0YPY7	MYLK2	Myosin light chain kinase 2
A0A8V0XI57	LOC100859645	Glutathione S-transferase alpha 3
Q9W6J2	GSTA4	Glutathione transferase
A0A3Q2TYR0	MYOZ3	Myozenin 3
E1C312	CAPN6	Calpain 6
A0A8V0YZK6	NMNAT1	Retinol binding protein 7
P27463	ALDH1A1	Aldehyde dehydrogenase 1A1
A0A8V1AEN9	DPP7	Dipeptidyl peptidase 7

## Data Availability

The raw data supporting the conclusions of this article will be made available by the authors on request.

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
