# Peer review of "Characterization of the Effects of Low-Sodium Salt Substitution on Sensory Quality, Protein Oxidation, and Hydrolysis of Air-Dried Chicken Meat and Its Molecular Mechanisms Based on Tandem Mass Tagging-Labeled Quantitative Proteomics"

_foods, 2024, doi:10.3390/foods13050737_

Round 1

Reviewer 1 Report

Comments and Suggestions for Authors

The manuscript “Characterization of the effects of low-sodium salt substitution on sensory quality, protein oxidation, and hydrolysis of air-dried chicken meat and its molecular mechanisms based on TMT-labeled quantitative proteomic” shows important results and relationships between molecular events and macro characteristics such as texture and sensory evaluation, discussing the findings in depth. I recommend correcting minor details described below.

​

Title: Write the meaning of TMT

Line 16: The percentages of the chlorides were calculated based on what?

Lines 88-95: Please try to explain all abbreviations.

Line 105: I do not consider that overall acceptability is evaluated with trained panelists, it is more appropriate with evaluation with consumers.

Lines 110-126: Explain with the verbs in the past tense.

Line 166: Explain the meanings of EM and EX.

Figure 2, 3, 4: The graphics should be large and clear for better visualization.

Line 360: Use the magnesium chloride formula.

Line 361: Write in full P.I., then the abbreviation.

Line 424: I think the word “destroyed” is very strong. I suggest changing.

Line 444: “meat products” is redundant.

Line 518: Hydrolysis index is repeated.

Comments on the Quality of English Language

Minor details can be corrected.

Author Response

Dear reviewer,

(Title: Write the meaning of TMT)

We appreciate for your valuable comment. We have modified TMT to its full name (tandem mass tagging).

(Line 16: The percentages of the chlorides were calculated based on what?)

We appreciate for your valuable comment. We have explained this in the reworded paragraph.All salt ratios were determined by calculating the ratio of the salt mass to the mass of the pickling solution.

(Lines 88-95: Please try to explain all abbreviations.)

We appreciate for your valuable comment. We have provided a corresponding explanation of the abbreviations that appear in the paragraph.

(Line 105: I do not consider that overall acceptability is evaluated with trained panelists, it is more appropriate with evaluation with consumers.)

Thanks for your comment. In this paper, overall acceptability represents a comprehensive judgment of subjects on the first five indicators of sensory ratings, but simply adding up the five ratings does not reflect this result, so overall acceptability was adopted for presentation. Therefore, We think this requires trained panelists.

(Lines 110-126: Explain with the verbs in the past tense.)

We appreciate for your valuable comment. We have revised this section to correct all the wrong tenses in the content.

(Line 166: Explain the meanings of EM and EX.)

We appreciate for your valuable comment. We have modified EM and EX to their full name ( emission wavelength and excitation wavelength)

(Figure 2, 3, 4: The graphics should be large and clear for better visualization.)

We appreciate for your valuable comment. We have enlarged these images so that the reader can clearly see the numbers and words on them.

(Line 360: Use the magnesium chloride formula.)

We appreciate for your valuable comment. We have changed magnesium chloride to MgCl2.

(Line 361: Write in full P.I., then the abbreviation.)

We appreciate for your valuable comment. We have changed P.I. to proteolysis index.

(Line 424: I think the word “destroyed” is very strong. I suggest changing.)

We appreciate for your valuable comment. We've changed "destroyed" to "changed".

(Line 444: “meat products” is redundant.)

We appreciate for your valuable comment. Duplicates of meat products have been removed.

(Line 518: Hydrolysis index is repeated.)

We appreciate for your valuable comment. Duplicates of hydrolysis index have been removed.

Thank you so much for your work to help me improve my article!

Reviewer 2 Report

Comments and Suggestions for Authors

Detailed comments are included in the review file attached below.

Comments on the Quality of English Language

The level of language is unsatisfactory. 
Extensive editing of the English language is required.
Detailed comments are included in the review file attached below.

Author Response

Dear reviewer,

(Title - titles do not end with full stops.)

We appreciate for your valuable comment. We've removed full stops that shouldn't be there.

(Abstract – linguistic errors, e.g. line 24 or line 18, 25 and in many other places  
in the manuscript (see comments on English level below). )

We appreciate for your valuable comment.  We have gone through lines 18, 24, and 25 and made changes.

(The Authors sometimes use ‘g’ and sometimes ‘gram’ for the unit of mass. )

We appreciate for your valuable comment. All grams have been denoted by g (except per gram when describing concepts which are still gram)

( Line 146 -147 or 201 - The Authors wrote: ‘the solution was centrifuged at 8000 g…’ – what does ‘g’ mean here? )

We appreciate for your valuable comment. Here g has all been replaced by × g, which is the unit of rotational speed.

(Subsection 2.8.2. – The title should be changed to ‘Chromatographic and mass spectrometry conditions’. Moreover, some values in this subsection are missing units in the description of these conditions. These should be introduced.)

We appreciate for your valuable comment. The title of Subsection 2.8.2. has been changed to 'Chromatographic and mass spectrometry conditions'. However, in reviewing the literature again, we found that I have no units missing from these data.

(A lot of abbreviations have been used in the manuscript, especially for chemical reagents but not only (e.g. BCA, DNTP or the others), which have not been explained before. Most of them will not be known to the readers, so they should be explained.)

We appreciate for your valuable comment. We have read through the entire article and explained all the abbreviations as they first appear.

( The title of  Figure 5 – was written in sentence form and needs to be rephrased.)

We appreciate for your valuable comment. We have modified The title of Figure 5, which is no longer in the form of a sentence but of a heading.

(English level.The level of language is unsatisfactory. The work contains some significant linguistic errors and needs significant improvement, especially in some parts of the work, including  the Introduction (e.g. lines 45, 49, 66-67) or Materials and Methods - there are errors in many places/lines, both grammatical, stylistic and in the nomenclature used in the technical language, and also in the Results’ section.  Furthermore, subsection 2.4. - was written in the form of instructions rather than descriptions, as is standard in scientific papers, and also contains grammatical errors.)

We appreciate for your valuable comment.  We have rewritten chapter 2.4 so that it illustrates the experimental steps concisely rather than describing them. For language issues throughout the text to have invited a professional to give revisions.

Thank you so much for all the help you have given me with your work.